# Reduced CXCL1 production by endogenous IL-37 expressing dendritic cells does not affect T cell activation

M. Kouwenberg[1☯], W. P. C. Pulskens[1☯], L. Diepeveen[1], M. Bakker-van Bebber[1], C. A. Dinarello[2,3], M. G. Netea[2], L. B. Hilbrands[1], J. van der Vlag[1]*

**1** Department of Nephrology, Radboud Institute of Molecular Life Sciences, Radboud University Medical Center, Nijmegen, The Netherlands, **2** Department of Internal Medicine and Radboud Center for Infectious Diseases, Radboud University Medical Center, Nijmegen, The Netherlands, **3** Department of Medicine, University of Colorado, Denver, Aurora, United States of America

☯ These authors contributed equally to this work.
* johan.vandervlag@radboudumc.nl

## Abstract

The dendritic cell (DC)-derived cytokine profile contributes to naive T cell differentiation, thereby directing the immune response. IL-37 is a cytokine with anti-inflammatory characteristics that has been demonstrated to induce tolerogenic properties in DC. In this study we aimed to evaluate the influence of IL-37 on DC–T cell interaction, with a special focus on the role of the chemokine CXCL1. DC were cultured from bone marrow of human IL-37 transgenic (hIL-37Tg) or WT mice. The phenotype of unstimulated and LPS-stimulated DC was analyzed (co-stimulatory molecules and MHCII by flow cytometry, cytokine profile by RT-PCR and ELISA), and T cell stimulatory capacity was assessed in mixed lymphocyte reaction. The role of CXCL1 in T cell activation was analyzed in T cell stimulation assays with anti-CD3 or allogeneic DC. The expression of the co-stimulatory molecules CD40, CD80 and CD86, and of MHCII in LPS-stimulated DC was not affected by endogenous expression of IL-37, whereas LPS-stimulated hIL-37Tg DC produced less CXCL1 compared to LPS-stimulated WT DC. T cell stimulatory capacity of LPS-matured hIL-37Tg DC was comparable to that of WT DC. Recombinant mouse CXCL1 did not increase T cell proliferation either alone or in combination with anti-CD3 or allogeneic DC, nor did CXCL1 affect the T cell production of interferon-γ and IL-17. Endogenous IL-37 expression does not affect mouse DC phenotype or subsequent T cell stimulatory capacity, despite a reduced CXCL1 production. In addition, we did not observe an effect of CXCL1 in T cell proliferation or differentiation.

## Introduction

Interleukin-37 (IL-37), a member of the IL-1 cytokine family, is a cytokine with strong anti-inflammatory properties [1]. In humans, IL-37 is expressed by different (immune) cell types upon stimulation, among which plasma cells, tissue macrophages, blood monocytes, dendritic cells, circulating B cells, natural killer cells, CD4+ T cells and regulatory T cells [2–4]. In

**Data Availability Statement:** All relevant data are within the paper and its Supporting Information files

**Funding:** JvdV; LH; MK: NWO ZonMw AGIKO 92003567, Radboud PhD program 2010. The funders had no role in study design, data collection and analysis, decision to publish, or preparation of the manuscript.

**Competing interests:** The authors have declared that no competing interests exist.

contrast to other IL-1 cytokine family members, IL-37 is not expressed in mice, although murine cells react to the human cytokine, suggesting the presence of IL-37 receptors and potentially a functional counterpart. Since the murine orthologue of IL-37 is unknown, the anti-inflammatory potential of IL-37 has been investigated in transgenic mice, overexpressing the human IL-37 (hIL-37Tg). In these hIL-37Tg mice, IL-37 appeared capable to reduce morbidity in a wide spectrum of experimental inflammatory [5–15] and auto-immune [16, 17] conditions.

The inhibitory effects of IL-37 on inflammatory processes involve both innate as well as adaptive immune cell responses. Dendritic cells (DC) are professional antigen presenting cells and crucial in naive T cell activation and differentiation, thereby shaping adaptive immunity. There are three levels of DC-derived signals that direct the T cell response and are involved in the balance between tolerance and immunity: the presentation of the MHC-antigen complex to the T cell receptor, the expression of co-stimulatory molecules, and the local cytokine milieu. Tolerogenic DC, characterized by a low expression of co-stimulatory molecules and an anti-inflammatory cytokine profile, can induce the generation of regulatory T cells [18], and are promising therapeutic targets in transplantation and autoimmune disease. By genetic manipulation or pharmacological interference during DC culture, attempts have been made to create tolerogenic DC producing low amounts of pro-inflammatory cytokines (e.g. IL-12) and high amounts of anti-inflammatory cytokines (e.g. IL-10, TGF-β). Based on the strong anti-inflammatory properties of IL-37, (increased) IL-37 expression in DC could result in a tolerogenic phenotype. Indeed, mouse DC expressing human IL-37 were shown to acquire a tolerogenic phenotype upon stimulation with LPS, with reduced expression of MHCII and co-stimulatory molecules, a reduced production of pro-inflammatory cytokines, and an increased production of anti-inflammatory cytokines [5, 11, 19]. Moreover, these DC reduced the allogeneic proliferative T cell response and increased the induction of regulatory T cells, compared to WT DC [11].

Based on aforementioned results we aimed to investigate the role of IL-37 in DC–T cell interaction in more detail. After observing a significantly lower production of CXCL1 in LPS-stimulated bone marrow derived hIL-37Tg DC as compared to WT DC, we focused especially on the role of CXCL1 in DC–T cell interaction.

## Material & methods

### Mice

Wild type C57Bl/6J and Balb/cAnNCrl male mice of 8–10 weeks old were obtained from Charles River (Maastricht, the Netherlands). Mice transgenic for human Interleukin-37 (hIL-37Tg) were created as described previously [5] and bred and housed under specific pathogen-free conditions in the animal facility of the Radboud university medical center in Nijmegen. Transgenic expression of IL-37 was confirmed by standard genotyping protocols on ear tissue samples. All mice received water and food *ad libitum*, and only age- and sex-matched mice were included in experiments. All animal experiments were carried out following the guidelines of local and national Animal Ethics Committees and after permission granted by the Animal Ethics Committee of the Radboud University Nijmegen (Permit Number 2011–024). Mice were sacrificed by cervical dislocation.

### Isolation and maturation of primary dendritic cells

For isolation and culture of primary cells, wild type and hIL-37Tg mice were sacrificed at age of 8–12 weeks. Isolation and culturing of bone marrow-derived DC in the presence of GM-CSF was performed as described in detail before [20–22]. DC were matured by addition

of prototype TLR-agonists (100 ng/ml LPS (from *Escherichia coli*, O111:B4), 1 μg/ml Pam3CSK, 1.3 μg/ml CpG ODN1826 or 1.3 μg/ml control ODN1826 (all from Invivogen, Carlsbad, USA)) for 24hrs. Control cells were treated with vehicle (PBS) to maintain an immature phenotype. Thereafter, cells were collected and either used in co-culture with T cells, or centrifuged in order to pellet cells for subsequent flow cytometry analysis and RNA isolation. Supernatant was harvested for subsequent ELISA measurements.

## Cell staining and flow cytometry

For cell staining and subsequent flow cytometric analysis, DC were harvested after 9 days of culture and subsequently stained, as previously described in detail [22, 23]. Cells were incubated with hamster-anti-mouse CD11c Alexa-647 antibodies (clone N418; diluted 1:100; AbD Serotec Kidlington, UK), in conjunction with either PE-labeled rat-anti-mouse CD40 IgG2a antibodies (clone FGK45.5; diluted 1:40; MACS, Miltenyi Biotec GmbH, Gladbach, Germany), or PE-labeled rat-anti-mouse CD86 IgG2b antibodies (clone PO3.1; diluted 1:80; eBioscience, Vienna, Austria), or PE-labeled Armenian hamster-anti-mouse CD80 IgG antibodies (clone 16-10A1, Biolegend, Fell, Germany), or PE-labeled rat-anti-mouse MHCII IgG2b antibodies (clone M5/114.15.2, eBioscience). The degree of staining was determined by flow cytometry (FC 500, Beckman Coulter, CA, USA) and analyzed using CFX software package (Beckman Coulter).

## Proliferation assay and mixed lymphocyte reaction

For both proliferation assay and mixed lymphocyte reaction (MLR), T cells (responder cells in MLRs) were obtained from Balb/c mice by mashing the spleen through a sterile, stainless 70 μm filter (Corning Inc., Corning, USA). Erythrocytes were lysed by Ammonium-Chloride-Potassium lysing buffer. T cells were 80–85% enriched from total splenocytes by depletion of MHCII-positive cells by magnetic cell sorting (MACS) using anti-MHCII magnetic beads and LS columns (Miltenyi Biotech GmbH, Bergisch Gladbach, Germany). Enriched T cells were intracellularly labeled with CFSE (Molecular Probes, Life Technologies Ltd, Paisley, UK) according to the manufacturer's instructions.

For T cell proliferation assay, 96 well round bottom plates (Corning Incorporated, Tewksbury, USA) were overnight coated with αCD3ε (clone 145-2C11, BD Biosciences, New Jersey, USA) and washed with sterile PBS. In each well $1 \times 10^5$ T cells were cultured in 200 μl medium supplemented with 10% FCS (at 37˚C, 95% humidity and 5% $CO_2$) for 4–6 days.

For MLRs, washed DC ($2.5 \times 10^4$ DC) were co-cultured with $1 \times 10^5$ T cells (at 37˚C, 95% humidity and 5% $CO_2$) for 4–6 days in 200 μl medium supplemented with 10% FCS in a 96 well round bottom plate (Corning Incorporated).

Proliferation of responder cells reflected by dilution of CFSE signal was measured by flow cytometry; all cells with reduced CFSE signal (compared to the peak of "static cells") were considered as proliferating cells. Culture supernatant was collected and stored at -20˚C for subsequent cytokine analysis. Where indicated, recombinant mouse CXCL1 (ProspecBio, Ness-Ziona, Israel) was added to the proliferation assay and/or MLR.

## RNA isolation and quantitative RT-PCR

Total RNA was isolated from cells using Trizol Reagent (Life Technologies; Bleiswijk, the Netherlands) according to the manufacturer's guidelines. All RNA samples were quantified by spectrophotometry and RNA was subsequently converted to cDNA using random hexamers and the RevertAid First Strand cDNA Synthesis kit (Thermo Scientific, Breda, The Netherlands). Gene expression was analyzed by real-time quantitative reverse-transcription (RT)-

**Table 1. Primer sequences used for quantitative RT-PCR.**

| Mouse Gene | Forward/reverse primer | Primer sequence 5'- 3' |
|---|---|---|
| HPRT | Forward | 5'-TCC TCC TCA GAC CGC TTT T-3' |
| | Reverse | 5'-CCT GGT TCA TCA TCG CTA ATC-3' |
| CXCL1 | Forward | 5'-ATA ATG CCC TTT TAC ATT CTT TAA CC-3' |
| | Reverse | 5'-AGT CCT TTG AAC GTC TCT GTC C-3' |
| TNF-α | Forward | 5'- CTGTAGCCCACGTCGTAGC -3' |
| | Reverse | 5'- TTGAGATCCATGCCGTTG -3' |
| IL-6 | Forward | 5'- TGTATCTCTCTGAAGGACT -3' |
| | Reverse | 5'-TCC TCC TCA GAC CGC TTT T-3' |
| IL-10 | Forward | 5'-GTG GAG CAG GTG AAG AGT GA-3' |
| | Reverse | 5'- TGC AGT TGA TGA AGA TGT-3' |
| IL-12 | Forward | 5'- GGAAGCACGGCAGCAGAATC -3' |
| | Reverse | 5'- AACTTGAGGGAGAAGTAGGAATGG -3' |
| IL-23 | Forward | 5'- TGGAGCAACTTCACACCTCC-3' |
| | Reverse | 5'- GGCAGCTATGGCCAAAAAGG-3' |

PCR performed on a Bio-Rad CFX96[TM] Real-Time qPCR, using SYBR Green Supermix (Roche Diagnostics; Mannheim, Germany). Specific gene expression was normalized to house-keeping gene (hypoxanthine-guanine phosphoribosyl transferase; HPRT) and analyzed using the $2^{-ddCt}$ method [24]. The mouse gene-specific primers are listed in Table 1.

## ELISA

CXCL1 (R&D Systems, catalog number DY453), IL-17 and interferon-γ (eBioscience, resp. catalog number 88-7472-88 and 88-7314-88) levels were measured using specific sandwich ELISA kits according to the manufacturer's protocol.

## Statistics

Differences between groups were analyzed using the Mann-Whitney U test with Graphpad (version 5.03 for Windows, GraphPad Software, San Diego, USA). Values are expressed as mean ± standard error of the mean (SEM), and a P-value ≤ 0.05 was considered as statistically significant.

## Results

### Endogenous expression of IL-37 does not affect the maturation of DC, but results in a reduced CXCL1 production

In order to evaluate the role of IL-37 in DC–T cell interaction, we first investigated the effect of endogenous IL-37 expression on DC maturation. Bone marrow derived DC from both hIL-37Tg and WT (C57Bl/6J) mice were cultured and 100 ng/ml LPS was added during final 24 hours of culture to induce maturation. Expression of co-stimulatory molecules (CD40, CD80, CD86) and MHCII expression on CD11c[+] DC was analyzed by flow cytometry. We found no effect of endogenous IL-37 on expression of co-stimulatory molecules or MHCII in both unstimulated (Fig 1A) and LPS-stimulated DC (Fig 1B). Similarly, upon stimulation with TLR2 (PAM3CSK) or TLR9 (ODN1826) agonists expression of co-stimulatory markers did not differ between hIL-37Tg and WT DC (S1 Fig).

Since the DC-derived cytokine profile dictates T cell differentiation, we subsequently analyzed the effect of IL-37 on cytokine mRNA expression profile in LPS-stimulated and

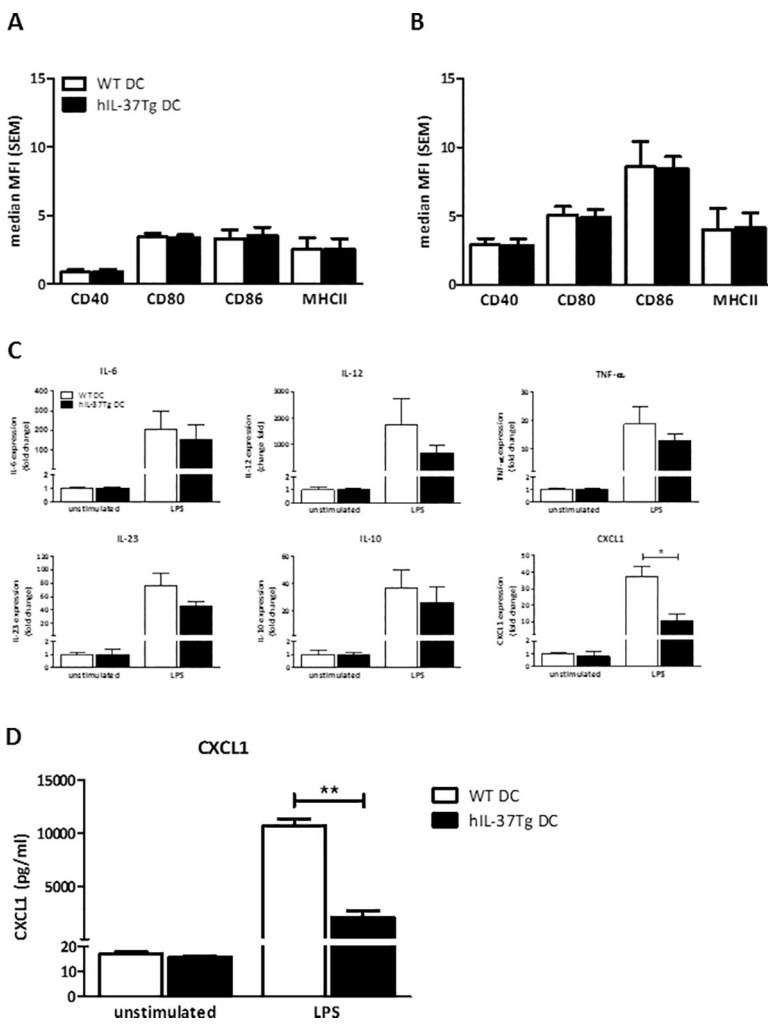

**Fig 1. Dendritic cell co-stimulatory molecule and MHCII expression is not affected by endogenous IL-37 expression.** LPS-matured hIL-37Tg DC produce significantly less CXCL1 compared to their WT counterparts: Bone marrow-derived DC (hIL-37Tg or C57Bl/6J) were differentiated within 9 days using GM-CSF. Expression of co-stimulatory molecules (CD40, CD80, CD86) and MHCII on unstimulated (A) and (100 ng/ml) LPS-stimulated (B) CD11c⁺ DC was subsequently analyzed by flow cytometry. RT-qPCR analysis of cytokine mRNA expression in hIL-37Tg and WT unstimulated and LPS-maturated DC with HPRT serving as housekeeping gene (C). Culture supernatants of hIL-37Tg LPS-stimulated and WT DC were analyzed for CXCL1 production (D). Data represent mean with SEM of 5 independent experiments. * p< 0.05, ** p< 0.001 (Mann Whitney test).

unstimulated DC. mRNA expression of IL-6, IL-10, IL-12, IL-23, and TNF-α was comparable between WT and hIL-37Tg DC upon LPS stimulation for 24h (Fig 1C). We found a strong decline in CXCL1 mRNA expression in LPS-maturated hIL-37Tg DC, compared to their WT counterparts (Fig 1C). In line, we found a significantly reduced CXCL1 protein production by LPS-stimulated hIL-37Tg DC compared to WT DC (Fig 1D).

## T cell stimulatory capacity of DC is not affected by endogenous IL-37 expression

We analyzed the T cell stimulatory capacity of hIL-37Tg DC in an allogeneic mixed lymphocyte reaction. DC were cultured during 9 days, in the presence or absence of LPS during the final 24 hours of culture. CFSE stained (Balb/c) MHCII-depleted splenocytes were co-cultured

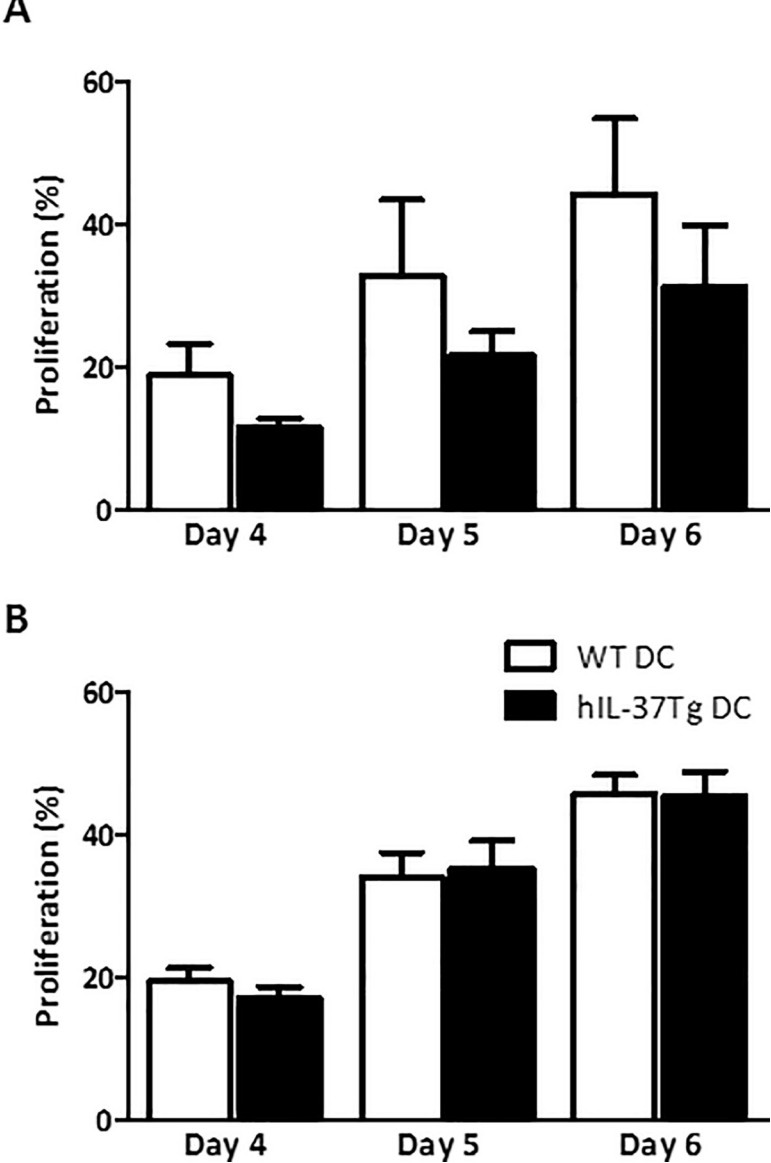

**Fig 2. Endogenous IL-37 expression in DC does not result in reduced T cell activation in MLR.** Stimulatory capacity of unstimulated (A) or (100 ng/ml) LPS-matured (B) WT (C57Bl/6J) or hIL-37Tg DC was tested in an allogeneic MLR. MHCII-depleted, CFSE stained Balb/c splenocytes were co-cultured with DC for 4–6 days. Proliferation was determined by measuring the dilution of the intracellular CFSE signal by flow cytometry. Data represent mean with SEM of 4 independent experiments.

for 4–6 days with unstimulated and LPS-stimulated DC, derived from WT or hIL-37Tg origin (C57Bl/6J). Fluorescent staining of MHCII-depleted splenocytes revealed 80–85% T cells (CD3+) and less than 2% B cells (CD19+). Proliferation was analyzed by measuring the dilution of the CFSE signal in flow cytometry.

We found no difference in proliferative response upon stimulation with hIL-37Tg or WT DC, either unstimulated (Fig 2A) or after stimulation with LPS (Fig 2B). Together this suggests that, despite reducing the production of CXCL1, endogenous expression of IL-37 does not affect the T cell stimulatory capacity of DC.

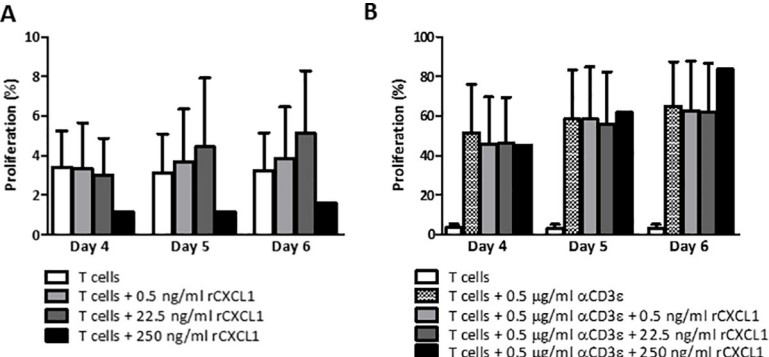

**Fig 3. Recombinant CXCL1 has no effect on CD3 stimulated T cell proliferation.** T cells were incubated for 4–6 days with (A) increasing concentrations of recombinant mouse CXCL1 (rCXCL1) (n = 3), or with (B) increasing concentrations of rCXCL1 in 0.5 µg/ml anti-CD3ε pre-coated plates (n = 3) (mean with SEM).

## Recombinant CXCL1 does not influence proliferation of T cells

Few chemokines have been reported capable to activate T cells [25, 26] and T cells have been shown to respond to CXCL1 [27–30]. To further evaluate whether CXCL1 contributes to T cell proliferation and differentiation, we performed a proliferation assay with MHCII-depleted splenocytes in the presence or absence of recombinant mouse CXCL1 (rCXCL1). The biological activity of rCXCL1 was confirmed in a neutrophil migration assay (data not shown).

After 4–6 days of incubation, no rCXCL1-induced proliferation could be observed, even in the presence of the highest rCXCL1 concentration (250 ng/ml, Fig 3A). Since rCXCL1 alone could be insufficient for T cell survival and activation, we used an additional assay in which rCXCL1 was added to anti-CD3 stimulation. MHCII-depleted splenocytes were cultured in anti-CD3ε pre-coated plates, to which increasing concentrations of rCXCL1 were added. While the stimulating effect of anti-CD3ε was clearly present, we did not observe an additional effect of rCXCL1 on T cell proliferation (Fig 3B).

## Unstimulated DC express little CXCR2, while T cells lack CXCR2 expression

Since we were not able to detect a direct proliferative effect of CXCL1 on T cells and since there are conflicting reports on the expression of CXCR2 (the receptor of CXCL1) on T cells, we investigated CXCR2 expression on Balb/c T cells. Since CXCL1 produced by the LPS-stimulated DC could have an autocrine effect on DC, thereby increasing their T cell stimulatory capacities, we also evaluated CXCR2 expression on DC.

We analyzed CXCR2 expression by flow cytometry on freshly isolated T cells (MHCII depleted splenocytes), unstimulated and LPS-stimulated DC,s and on granulocytes as a positive control. Granulocytes isolated from peripheral blood clearly express CXCR2 (Fig 4A), while T cells lack CXCR2 cell surface expression (Fig 4B). Unstimulated DC express low levels of CXCR2 (Fig 4C), which disappears upon LPS stimulation (Fig 4D).

In summary, unstimulated DC express low levels of CXCR2 on their cell surface, while T cells lack CXCR2 expression.

## Recombinant CXCL1 has no effect on DC stimulated T cell proliferation or differentiation

In the absence of CXCR2 on T cells, a direct effect of rCXCL1 on T cell activation or proliferation seems unlikely. DC-derived CXCL1 could exert an autocrine effect on the DC, affecting T

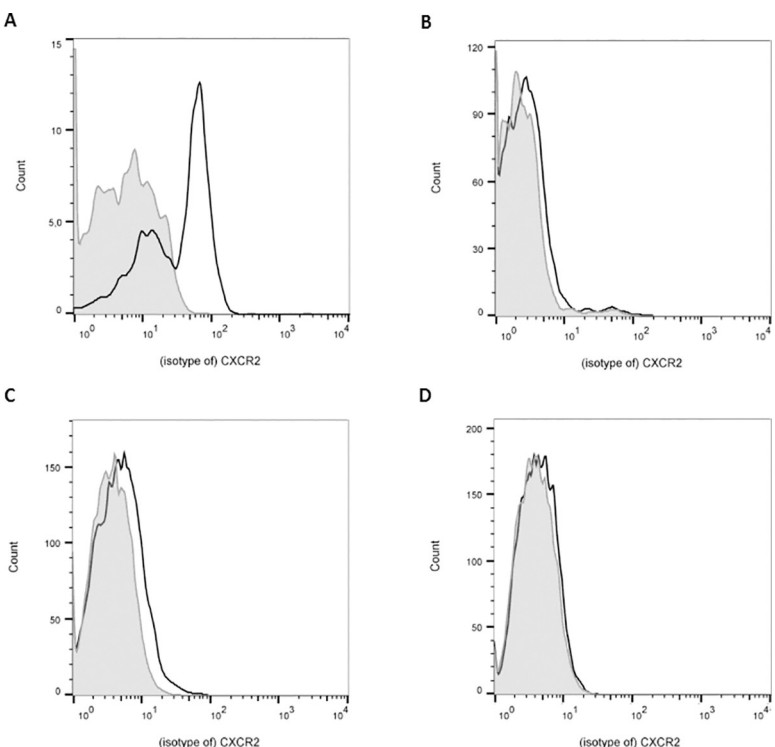

**Fig 4. CXCR2 expression on DC and T cells.** CXCR2 expression was analyzed by flow cytometry. Granulocytes (Gr1 +; A), Balb/c T cells (CD3+; B) and unstimulated (C) and LPS-stimulated DC (CD11c+; D) were stained with anti-CXCR2 (black line) or isotype control (grey area). Granulocytes stained positive for CXCR2, while expression was low in unstimulated DC and absent in LPS-stimulated DC and T cells. Representative histograms of 4 independent experiments.

cell functioning indirectly. Therefore, we performed a MLR assay with unstimulated DC as stimulators in the presence or absence of rCXCL1. The stimulating effect of DC on T cell proliferation was not enhanced by adding rCXCL1 (Fig 5A).

While human CXCR2-transfected T cells produced interferon-γ (IFN-γ) upon incubation with CXCL1, naïve mouse CD4+ T cells have been reported to produce IL-17 upon stimulation by CXCL1 [31–33]. Therefore, we measured IFN-γ and IL-17 in culture supernatant, but did not find any change in either IFN-γ or IL-17 production related to the presence of rCXCL1 (Fig 5B and 5C). In summary, CXCL1 does not seem to contribute to DC-induced proliferation of T cells nor production of IFN-γ and IL-17.

## Discussion

DC are key players in adaptive immunity and direct the skewing of T cell differentiation, mainly by skewing their cytokine profile. The anti-inflammatory cytokine IL-37 has been shown to mitigate immune responses in experimental inflammatory conditions [5–15] and autoimmune diseases [16, 17]. Since these effects could be favorable in the setting of organ transplantation, we aimed to further unravel the role of IL-37 in DC–T cell interaction, focusing on the effects of IL-37 on DC maturation and T cell stimulatory capacity. We found that endogenous expression of IL-37 does not affect the phenotype of LPS-matured DC, although we found a decline in CXCL1 production by IL-37 expressing LPS-matured DC compared to WT DC.

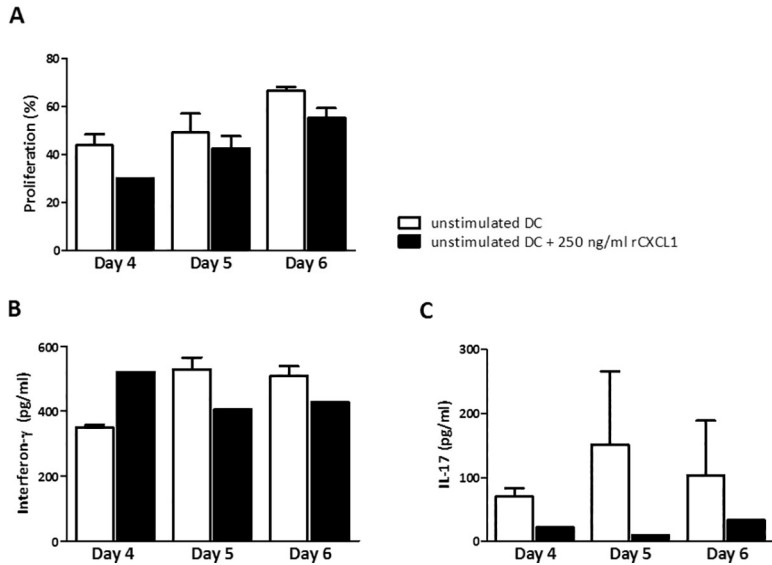

**Fig 5. Recombinant CXCL1 has no effect on DC induced T cell stimulation.** Proliferative response of T cells (MHCII depleted splenocytes, Balb/c) co-cultured with unstimulated DC (C57Bl/6J) in the absence or presence of 250 ng/ml rCXCL1 (n = 2). Interferon-γ (B) and IL-17 (C) concentration in culture supernatants collected at day 4–6 of culture (mean with SEM).

The reduced production of CXCL1 induced by LPS in hIL-37Tg DC was not accompanied by a reduced cell surface expression of co-stimulatory molecules, a lower mRNA expression of inflammatory cytokines, or reduced ability to stimulate allogeneic T cells, which is in contrast with findings by others [5, 11, 34, 35]. Several differences in experimental setup can partly explain these conflicting results. We cultured DC from bone marrow of hIL-37Tg mice, while others [34, 35] added recombinant IL-37 to WT DC in culture. Nold et al. analyzed splenic DC of hIL-37Tg mice 24 hours after LPS injection in-vivo [5]. Luo et al. used a similar experimental setup as we did for culturing hIL-37Tg DC, but found significantly reduced expression of MHCII and CD40 in LPS-matured hIL-37Tg DC as compared to WT DC, accompanied by a reduced production of pro-inflammatory cytokines and increased secretion of IL-10 [11]. A clear explanation for these conflicting data is lacking.

CXCL1, also known as keratinocyte chemoattractant and Gro-α, is a chemokine that belongs to the CXC chemokine family and is expressed both in humans and mice. CXCL1 plays a role in inflammation, angiogenesis, wound healing, and cancer development. Little is known about the direct effect of DC-derived CXCL1 on T cell functioning. However, intragraft expression of CXCL1 has been associated with accelerated rejection and rejection could be promoted by CXCL1 [36, 37]. Although CXCL1 is a known chemoattractant for neutrophils, it was suggested to (indirectly) attract T cells as well, thereby contributing to accelerated rejection [36]. The clear inhibitory effect of IL-37 on CXCL1 production has been described before. In LPS-induced inflammation models, as well as an arthritis model and in macrophages, CXCL1 production is strongly decreased by IL-37 [5, 8, 16, 38]. The reduced CXCL1 production could be explained by the inhibitory effects of IL-37 expression on Nuclear Factor kappa B (NFκB) activity, a transcription factor driving CXCL1 transcription [6, 39, 40]. Indeed, expression of NFκB is reduced in hIL-37Tg DC [19]. Along with reduced TLR4 expression IL-37 appears to affect NFκB expression via increased IL-1 receptor 8 expression. Based on its nuclear translocation, IL-37 is thought to directly affect gene transcription as well. The intracellular precursor form of IL-37 needs caspase-1 processing in order to translocate to

the nucleus [41] and upon caspase-1 inhibition the anti-inflammatory properties of IL-37 were abolished [4].

The reduced production of CXCL1 under influence of IL-37 combined with previously published data on decreased T cell stimulation by IL-37 expressing DC [11], suggested that CXCL1 may be involved in DC induced T cell stimulation. We were not able to detect expression of CXCR2, the receptor for CXCL1, on the surface of T cells by flow cytometry, which makes a direct effect of CXCL1 on T cells resulting in T cell proliferation less likely. CXCR2 gene expression has been described in T cells from older mice (18–22 months of age) and mammary tumor bearing mice [42–45], while we used young (aged 8–12 weeks) and healthy mice. Theoretically, CXCR2 could be expressed upon activation of T cells, though we found no effect of CXCL1 on T cells activated by anti-CD3 or DC either. Altogether, we found no evidence for a direct role of DC-derived CXCL1 on murine T cells.

Subsequently, we hypothesized that there was an autocrine function for DC-derived CXCL1, thereby potentiating the stimulating effects of DC on naïve T cells. A proportion of mouse DCs in the circulatory system express CXCR2 [46]. While others could not demonstrate CXCR2 expression on DC of different mouse strains [47, 48], it has been shown on human monocyte derived DC [49, 50]. CXCL1 has been described to affect DC maturation and cytokine production [51].We found low expression levels of CXCR2 on the surface of unstimulated DC which disappeared after exposure to LPS. However, adding rCXCL1 to a co-culture of unstimulated DC and T cells did not enhance T cell proliferation. We therefore conclude that a pro-inflammatory, autocrine effect of DC-derived CXCL1 is unlikely. This is supported by the finding that incubation of human monocyte derived DC with CXCL1 prior to LPS exposure resulted in reduced production of the pro-inflammatory cytokine IL-12p70 [51].

In this study we addressed the hypothesis that IL-37 can induce a tolerogenic properties in DC and thereby impacts T cell adaptive immunity. We could not confirm earlier findings of reduced expression of MHCII and CD40 and the switch to a tolerogenic cytokine profile, but found reduced CXCL1 production in LPS-matured hIL37-Tg DC. We speculate that DC-derived CXCL1 contributes to T cell activation, but found no evidence for a role of CXCL1 in T cell activation by DC or other stimuli.

## Supporting information

**S1 Fig. Expression of co-stimulatory molecules CD40 and CD86 in PAM3 and ODN stimulated bone marrow derived dendritic cells of hIL37Tg and WT mice.** Bone marrow-derived DC (hIL-37Tg or C57Bl/6J) were differentiated within 9 days using GM-CSF. Expression of co-stimulatory molecules (CD40, CD86) on 1 μg/ml Pam$_3$CSK or 1.3 μg/ml CpG ODN1826 stimulated CD11c$^+$ DC was subsequently analyzed by flow cytometry.
(TIF)

## Author Contributions

**Conceptualization:** M. Bakker-van Bebber, L. B. Hilbrands, J. van der Vlag.

**Data curation:** M. Kouwenberg, W. P. C. Pulskens.

**Formal analysis:** M. Kouwenberg, W. P. C. Pulskens, L. Diepeveen, L. B. Hilbrands, J. van der Vlag.

**Funding acquisition:** L. B. Hilbrands, J. van der Vlag.

**Investigation:** M. Kouwenberg, W. P. C. Pulskens, L. Diepeveen, M. Bakker-van Bebber.

**Methodology:** M. Kouwenberg, W. P. C. Pulskens, C. A. Dinarello, L. B. Hilbrands, J. van der Vlag.

**Project administration:** M. Kouwenberg, W. P. C. Pulskens, L. B. Hilbrands, J. van der Vlag.

**Resources:** M. G. Netea.

**Supervision:** L. B. Hilbrands, J. van der Vlag.

**Validation:** M. Kouwenberg, W. P. C. Pulskens, L. B. Hilbrands, J. van der Vlag.

**Visualization:** M. Kouwenberg, W. P. C. Pulskens, L. B. Hilbrands, J. van der Vlag.

**Writing – original draft:** M. Kouwenberg, W. P. C. Pulskens.

**Writing – review & editing:** W. P. C. Pulskens, L. Diepeveen, M. G. Netea, L. B. Hilbrands, J. van der Vlag.

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
