## [Decision Letter · Decision Letter 0]

3 Mar 2021

PONE-D-20-39909

Reduced CXCL1 production by endogenous IL-37 expressing dendritic cells does not affect T cell activation

PLOS ONE

Dear Dr. van der Vlag,

Thank you for submitting your manuscript to PLOS ONE. After careful consideration, we feel that it has merit but does not fully meet PLOS ONE’s publication criteria as it currently stands. Therefore, we invite you to submit a revised version of the manuscript that addresses the points raised during the review process.

Reviewers have provided comments to improve the text. Also address the question on 'novelty' in the discussion

We look forward to receiving your revised manuscript.

Kind regards,

Jagadeesh Bayry, DVM, PhD, HDR

Academic Editor

PLOS ONE

Journal Requirements:

3. Please provide the catalog numbers and sources of the sandwich ELISA kits.

Reviewers' comments:

Reviewer's Responses to Questions

**Comments to the Author**

1. Is the manuscript technically sound, and do the data support the conclusions?

Reviewer #1: Yes

Reviewer #2: Yes

2. Has the statistical analysis been performed appropriately and rigorously? 

Reviewer #1: Yes

Reviewer #2: Yes

3. Have the authors made all data underlying the findings in their manuscript fully available?

Reviewer #1: Yes

Reviewer #2: Yes

4. Is the manuscript presented in an intelligible fashion and written in standard English?

Reviewer #1: Yes

Reviewer #2: Yes

5. Review Comments to the Author

Reviewer #1: This study provides evidence that the endogenous expression of IL-37 does not affect the phenotype of LPS-matured dendritic cells (DC) while a decrease in CXCL1 production by IL-37 expressing is observed in LPS-matured DC when compared to WT DC. Addition This study provides evidence that ally, CXCL1 is not involved in T cell activation by DC or other stimuli. Overall, this study is very interesting and the manuscript is well written and clearly presented.

Major concern,

It is not clear why the authors focused only on the effect of the chemoattractant CXCL1 on T cell activation? Did the authors assess the effect of CXCL1 on B cell activation? please discuss.

Minor concern

The Y- and X-axis of Fig. 5C are not clear.

Reviewer #2: Kouwenberg et al. present a study looking at the role of the anti-inflammatory IL-37 expression in the production of the chemokine, CXCL1, from dendritic cells. They found that IL-37 dampened CXCL1 expression and that the changes in CXCL1 expression did not affect T cell activation. Overall, the manuscript is easy to follow and read. Though the manuscript does show a negative results, I firmly believe negative results play a significant role in our scientific knowledge. Unfortunately the finding in Kouwenberg et al. are not novel. A quick literature search will find that CXCL1 is a chemokine that is not involved in T cell activation, due in part to T cells low/absence of the CXCL1 target receptor CXCR2 expression. CXCL1 is commonly found to affect the innate immune response and neutrophil and not T cell chemotaxis. A more relevant study would have looked at how IL-37 expression affects CXCL1 and how that in turn supports either neutrophil activation or chemotaxis.

6. PLOS authors have the option to publish the peer review history of their article (what does this mean?). If published, this will include your full peer review and any attached files.

Reviewer #1: **Yes: **Samir Jawhara

Reviewer #2: No

---

## [Author Response · Author response to Decision Letter 0]

6 Apr 2021

Reviewer #1: 

“This study provides evidence that the endogenous expression of IL-37 does not affect the phenotype of LPS-matured dendritic cells (DC) while a decrease in CXCL1 production by IL-37 expressing is observed in LPS-matured DC when compared to WT DC. Addition This study provides evidence that ally, CXCL1 is not involved in T cell activation by DC or other stimuli. Overall, this study is very interesting and the manuscript is well written and clearly presented.”

We thank the reviewer for the very positive comments.

Major concern,

“It is not clear why the authors focused only on the effect of the chemoattractant CXCL1 on T cell activation? Did the authors assess the effect of CXCL1 on B cell activation? please discuss.”

In our manuscript, we focus on the role of IL-37 expression on dendritic cell (DC) functioning and subsequent DC-induced T cell activation. We found that LPS-stimulated hTgIL-37 mouse DC produce significantly less CXCL1 than their WT counterparts. It has been reported that IL-37 expressing DC display reduced T cell activation (1, 2), and that T cells can respond to CXCL1 (3-8). We therefore hypothesized that the reduced T cell activating capacity of IL-37 expressing DC is mediated by reduced CXCL1 production. To our knowledge there are no reports clearly showing responses of B cells to CXCL1. Therefore, we focused on the potential role of DC-derived CXCL1 in T cell activation.

To further clarify our further focus on the role of CXCL1 in T cell activation, we included these details to the manuscript in the result section (line 212-213).

Minor concern

The Y- and X-axis of Fig. 5C are not clear.

We are a bit puzzled regarding this comment. Figure 5C depicts IL-17 levels in supernatant of MHC-II depleted splenocytes (Balb/c) co-cultured with unstimulated DC (C57Bl/6J) in the absence (white) or presence (black) of 250 ng/ml rCXCL1. Please let us know whether we have overlooked something.

Reviewer #2: 

Kouwenberg et al. present a study looking at the role of the anti-inflammatory IL-37 expression in the production of the chemokine, CXCL1, from dendritic cells. They found that IL-37 dampened CXCL1 expression and that the changes in CXCL1 expression did not affect T cell activation. Overall, the manuscript is easy to follow and read. Though the manuscript does show a negative results, I firmly believe negative results play a significant role in our scientific knowledge. 

We completely agree that “negative” results should be published in the public domain.

Unfortunately the finding in Kouwenberg et al. are not novel. A quick literature search will find that CXCL1 is a chemokine that is not involved in T cell activation, due in part to T cells low/absence of the CXCL1 target receptor CXCR2 expression. CXCL1 is commonly found to affect the innate immune response and neutrophil and not T cell chemotaxis. A more relevant study would have looked at how IL-37 expression affects CXCL1 and how that in turn supports either neutrophil activation or chemotaxis.

We appreciate this comment and we agree that investigating the consequences of IL-37 expression induced changes in CXCL1 production on neutrophil chemotaxis would be interesting.

However, as part of our research program on tolerizing DC for the prevention of allograft rejection, we were especially interested in a potential role of CXCL1 in the DC-T cell interaction. Notably, there is literature that describes responses of T cells to CXCL1 (3-8). In addition, recombinant human CXCL1 has been described as potent chemoattractant for freshly isolated human T cells (8, 9). CD4+ and CD8+ T cells were attracted to an equal extent, with predominant migration of the CD45+ memory T cell subset (9). In mice, CXCR2 expression has been detected on lymphocytes, as described in the manuscript. Furthermore, a possible autocrine effect of DC-derived CXCL1 was supported by the observation that a proportion of murine circulating CD11c+ DC express CXCR2 (10), and CXCL1 had been described to affect DC maturation and cytokine production (11).

Based on these published data and our findings of reduced production of CXCL1 by IL-37 expressing DC, we focused on the effects of DC-derived CXCL1 on T cell activation. Since there are no detailed studies on DC-derived CXCL1 on T cell activation, we think our data set, although “negative”, is novel. 

We further revised the discussion section of the manuscript (line 313-317) by adding the above mentioned details. 

References

1. Luo Y, Cai X, Liu S, Wang S, Nold-Petry CA, Nold MF, et al. Suppression of antigen-specific adaptive immunity by IL-37 via induction of tolerogenic dendritic cells. Proceedings of the National Academy of Sciences of the United States of America. 2014;111(42):15178-83.

2. Nold MF, Nold-Petry CA, Zepp JA, Palmer BE, Bufler P, Dinarello CA. IL-37 is a fundamental inhibitor of innate immunity. Nature immunology. 2010;11(11):1014-22.

3. Khaw YM, Tierney A, Cunningham C, Soto-Díaz K, Kang E, Steelman AJ, et al. Astrocytes lure CXCR2-expressing CD4(+) T cells to gray matter via TAK1-mediated chemokine production in a mouse model of multiple sclerosis. Proceedings of the National Academy of Sciences of the United States of America. 2021;118(8).

4. Liu YJ, Guo DW, Tian L, Shang DS, Zhao WD, Li B, et al. Peripheral T cells derived from Alzheimer's disease patients overexpress CXCR2 contributing to its transendothelial migration, which is microglial TNF-alpha-dependent. Neurobiology of aging. 2010;31(2):175-88.

5. Takata H, Tomiyama H, Fujiwara M, Kobayashi N, Takiguchi M. Cutting edge: expression of chemokine receptor CXCR1 on human effector CD8+ T cells. Journal of immunology (Baltimore, Md : 1950). 2004;173(4):2231-5.

6. Rajaei T, Farajifard H, Rezaee SA, Azarpazhooh MR, Mahmoudi M, Valizadeh N, et al. Different roles of CXCR1 and CXCR2 in HTLV-1 carriers and HTLV-1-associated myelopathy/tropical spastic paraparesis (HAM/TSP) patients. Med Microbiol Immunol. 2019;208(5):641-50.

7. Li YM, Liu ZY, Wang JC, Yu JM, Li ZC, Yang HJ, et al. Receptor-Interacting Protein Kinase 3 Deficiency Recruits Myeloid-Derived Suppressor Cells to Hepatocellular Carcinoma Through the Chemokine (C-X-C Motif) Ligand 1-Chemokine (C-X-C Motif) Receptor 2 Axis. Hepatology. 2019;70(5):1564-81.

8. Chen P, Yi Z, Zhang W, Klotman ME, Chen BK. HIV infection-induced transcriptional program in renal tubular epithelial cells activates a CXCR2-driven CD4+ T-cell chemotactic response. AIDS (London, England). 2016;30(12):1877-88.

9. Jinquan T, Frydenberg J, Mukaida N, Bonde J, Larsen CG, Matsushima K, et al. Recombinant human growth-regulated oncogene-alpha induces T lymphocyte chemotaxis. A process regulated via IL-8 receptors by IFN-gamma, TNF-alpha, IL-4, IL-10, and IL-13. Journal of immunology (Baltimore, Md : 1950). 1995;155(11):5359-68.

10. Katoh H, Wang D, Daikoku T, Sun H, Dey SK, Dubois RN. CXCR2-expressing myeloid-derived suppressor cells are essential to promote colitis-associated tumorigenesis. Cancer Cell. 2013;24(5):631-44.

11. Michielsen AJ, Hogan AE, Marry J, Tosetto M, Cox F, Hyland JM, et al. Tumour tissue microenvironment can inhibit dendritic cell maturation in colorectal cancer. PloS one. 2011;6(11):e27944.

---

## [Decision Letter · Decision Letter 1]

4 May 2021

Reduced CXCL1 production by endogenous IL-37 expressing dendritic cells does not affect T cell activation

PONE-D-20-39909R1

Dear Dr. van der Vlag,

We’re pleased to inform you that your manuscript has been judged scientifically suitable for publication and will be formally accepted for publication once it meets all outstanding technical requirements.

Kind regards,

Jagadeesh Bayry, DVM, PhD, HDR

Academic Editor

PLOS ONE

Additional Editor Comments (optional):

Reviewers' comments:

Reviewer's Responses to Questions

**Comments to the Author**

1. If the authors have adequately addressed your comments raised in a previous round of review and you feel that this manuscript is now acceptable for publication, you may indicate that here to bypass the “Comments to the Author” section, enter your conflict of interest statement in the “Confidential to Editor” section, and submit your "Accept" recommendation.

Reviewer #1: All comments have been addressed

Reviewer #2: All comments have been addressed

2. Is the manuscript technically sound, and do the data support the conclusions?

Reviewer #1: Yes

Reviewer #2: Yes

3. Has the statistical analysis been performed appropriately and rigorously? 

Reviewer #1: Yes

Reviewer #2: Yes

4. Have the authors made all data underlying the findings in their manuscript fully available?

Reviewer #1: Yes

Reviewer #2: Yes

5. Is the manuscript presented in an intelligible fashion and written in standard English?

Reviewer #1: Yes

Reviewer #2: Yes

6. Review Comments to the Author

Reviewer #1: The authors have addressed most of the concerns that I had with this new submission. I dont have any additional comments for the authors.

Reviewer #2: The authors of the manuscript, "Reduced CXCL1 production by endogenous IL-37 expressing dendritic cells does not affect T cell activation" have adequately addressed my concerns about the manuscript.

7. PLOS authors have the option to publish the peer review history of their article (what does this mean?). If published, this will include your full peer review and any attached files.

Reviewer #1: **Yes: **Samir Jawhara

Reviewer #2: No

---

## [Editor Report · Acceptance letter]

14 May 2021

PONE-D-20-39909R1 

Reduced CXCL1 production by endogenous IL-37 expressing dendritic cells does not affect T cell activation 

Dear Dr. van der Vlag:

I'm pleased to inform you that your manuscript has been deemed suitable for publication in PLOS ONE. Congratulations! Your manuscript is now with our production department. 

Kind regards, 

on behalf of

Dr. Jagadeesh Bayry 

Academic Editor

PLOS ONE